# Motherhood in Europe: An Examination of Parental Leave Regulations and Breastfeeding Policy Influences on Breastfeeding Initiation and Duration

**Karen Vanderlinden** [1], **Veerle Buffel** [2], **Bart Van de Putte** [1] and **Sarah Van de Velde** [2,*]

[1] Department of Sociology, Health and Demographic Research, Ghent University, 9000 Gent, Belgium; Karen.Vanderlinden@UGent.be (K.V.); bart.vandeputte@ugent.be (B.V.d.P.)

[2] Department of Sociology, Centre for Population, Family and Health, University of Antwerp, 2000 Antwerp, Belgium; veerle.buffel@uantwerpen.be

[*] Correspondence: sarah.vandevelde@uantwerpen.be

**Abstract:** This study examines how European variation in breastfeeding initiation and duration rates is related to the presence of baby-friendly hospitals, the international code of marketing of breast-milk substitutes, and different constellations of maternal, paternal, and parental leave. We use Eurobarometer data (2005) to compare initiation and duration levels across 21 European countries within a multilevel regression framework. We find that countries play a significant role in determining breastfeeding through their different social policies. Breastfeeding practices across different leave regulation models differ substantially. We conclude that ongoing changes in paid maternity and parental leave length combined with uptake flexibility and paternal involvement help determine breastfeeding rates and should put infant feeding issues on governmental policy agendas across European countries.

**Keywords:** breastfeeding initiation; breastfeeding duration; social policy in Europe; maternity; paternity; and parental leave; baby-friendly hospitals; international code of marketing of breast-milk substitutes

## 1. Background

Europe has significant levels of cross-national variation in the initiation and duration of breastfeeding, as shown in several descriptive studies (Cattaneo et al. 2005; Victora et al. 2016; Yngve and Sjöström 2001). In Northern European countries, most women initiate breastfeeding; a substantial number also continue breastfeeding beyond the initial weeks following childbirth. In countries like Poland, the Czech Republic, and Bulgaria, prevalence rates of breastfeeding initiation are also high, but duration rates are substantially lower. Other countries, such as the UK, France, and Belgium, have low initiation and duration rates.

Despite this cross-national variation in breastfeeding practices, available research on the topic has mainly identified individual-level correlates of breastfeeding initiation (from now on BFI) and breastfeeding duration (from now on BFD). Lower likelihood of breastfeeding is related to a number of socio-medical factors such as lack of knowledge of (the benefits of) breastfeeding, technique difficulty (e.g., latching), medical difficulties (e.g., mastitis, cracked nipples, or thrush infections), or a mother's worry that her child is not getting enough milk (Baker and Milligan 2008). A mother's confidence in the ability to breastfeed increases the likelihood of initiation, while experiencing breastfeeding problems increases the likelihood of (early) cessation (Taveras et al. 2003). Additionally, both initiation and duration success are linked to women's social position (Colodro-Conde et al. 2011; Senarath et al. 2010). Women from a vulnerable socioeconomic background, as well as women with

less social support, are less likely to initiate breastfeeding and to opt out of breastfeeding earlier (Barona-Vilar et al. 2009; Colodro-Conde et al. 2011; Earle 2002; Heck et al. 2006). Mothers are more likely to continue breastfeeding if they receive encouragement and support from medical professionals (Taveras et al. 2003; Ekström et al. 2003; Swanson and Power 2005; Barona-Vilar et al. 2009) or from significant others (Kools et al. 2006; Ekström et al. 2003; Swanson and Power 2005). Many studies find that returning to work is the strongest predictor of breastfeeding discontinuation (Taveras et al. 2003; Kools et al. 2006; Barona-Vilar et al. 2009; Rippeyoung and Noonan 2012). Returning to work acts as an important barrier to breastfeeding duration, since pumping, using teats, and freezing milk is a time-consuming activity, and colleagues' reaction often discourages this process (Hawkins et al. 2008). Reinforcing factors, such as workplace support, maternity leave benefits, and media advocacy, strongly affect the success of continued breastfeeding among working mothers (Yngve and Sjöström 2001; Senarath et al. 2010). While breastfeeding may help reinforce the maternal identity (Lee 2008), some women also express a desire to reestablish their identities as separate individuals and as "non-mothers" by discontinuing breastfeeding (Earle 2002).

However, the general cultural and institutional context influences many of these factors. From a socio-ecological perspective, policy is one of the outer layers influencing breastfeeding knowledge, beliefs, and skills (Bentley et al. 2003; Hodgson 1986), with several organizational, community-level, and interpersonal influences in between (see Figure 1). What the socio-ecological framework doesn't incorporate is how gender inherently connects to breastfeeding. Gender operates at three levels (Risman 2004): *personal, structural,* and *cultural.* First, on a *personal* level, gender is part of a person's identity and shapes everyday experiences. An example of this is how someone experiences being a mother and the embodied experience of breastfeeding itself (i.e., sexualization versus the maternal function of the body) as a performative action in front of others in public and/or private spheres (Stearns 1999; Blum 2000; Fox and Neiterman 2015). Second, on the *structural* level, in how societies organize and categorize desired qualities and jobs. For example, if a society values taking time off to perform care work less than continuing to work and outsourcing care responsibilities, then leave systems will reflect this categorization. Moreover, last, on a *cultural* level, gender operates as a basis for prevailing norms and values and as a basis for desired behavior within society, which customs, rules, language, media, political and economic systems—and policies—reflect. Leave systems that are generous and compensate both mothers and fathers, for example, indicate an emphasis on creating gender equality and gender norms that value parental childcare. The personal, structural, and cultural levels interact with and reinforce each other, which means the policy must consider all three to facilitate change. In sum, gender permeates all socio-ecological levels through society's conceptualization of gender equality, categories, and ideology (Blum and Vandewater 1993). When policy explicitly values parental childcare, parenthood is (temporarily) prioritized over workplace demands and can be paramount in providing a successful work-life balance.

To date, only a handful of studies have examined how context explains cross-national variation in breastfeeding. A study by Galtry (2003) described the work-family reconciliation policies of Ireland, the UK, and Sweden. It related the higher breastfeeding rates in the Swedish model to Sweden's well-developed parental leave scheme. Yngve and Sjöström (2001) examined the degree to which EU member states comply with the WHO (World Health Organization)/UNICEF (United Nations Children's Fund) recommendations concerning breastfeeding (WHO 2003) and related these recommendations to cross-national differences in breastfeeding rates. Nonetheless, they conclude that breastfeeding rates are difficult to compare across countries because there is little consistency in reporting breastfeeding statistics across countries, and there are varying definitions of breastfeeding. Cattaneo et al. (2005) asked health personnel in 18 European countries about adherence to the Baby-friendly hospital initiative, adherence to the International Code of Marketing of Breast-milk Substitutes, the degree to which volunteer groups were active in breastfeeding support, and the rates of both exclusive and complementary breastfeeding. They relate the high degree of variation in breastfeeding initiation and duration to adherence to these breastfeeding policies. In line with Yngve and Sjöstrom, however,

they concluded that cross-national comparisons were difficult due to data limitations. More recent studies confirm previous findings regarding comparable design difficulties in the European landscape (Bosi et al. 2015; Lubold 2017). Bosi et al. (2015) emphasize how rates vary substantially across the European region, and even while many European countries have specific policies in place, there is a failure in translating this into higher breastfeeding rates across the board. The study by Lubold (2017) applied a comparable research design across 18 high-income countries. It found that women were more likely to initiate breastfeeding in countries that have a high percentage of women in parliament, a low national cesarean section rate, and either low public spending on family benefits, high rates of maternity leave, or high rates of women working part-time. It also found that low national adherence to the baby-friendly hospital initiative was related to low breastfeeding initiation. Unfortunately, the available cross-national studies on BFI and BFD are limited in several ways. First, these studies estimated cross-national differences using either convenience samples or meta-analyses of data from a diverse set of studies using different definitions of BFI and BFD and different sampling designs and populations. Moreover, they included only macro-level indicators. Consequently, the conclusions are subject to an ecological fallacy, making it difficult to identify whether these policy effects have an independent effect on breastfeeding or whether they simply reflect differences in the composition of the population of mothers across countries.

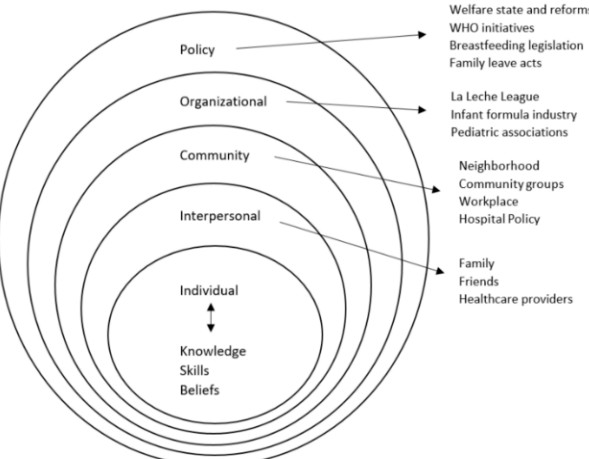

**Figure 1.** A social-ecological framework for breastfeeding. Source: An adaption from the social-ecological framework in Bentley et al. (2003).

The current study builds upon and extends these cross-national studies on BFI and BFD within a multilevel framework while also overcoming the shortcomings of these studies. We aim at making a contribution to different streams of research: the literature on breastfeeding and public health, earlier research on parental leave and policy effects, the socioeconomic determinants of health-related behaviors, research on parenthood. We use data from the Eurobarometer 2005, which collected information on several socioeconomic and demographic indicators as well as on BFI and BFD from the general population in 21 European countries. Our main research question is how countries' social policies shape individual breastfeeding outcomes while also considering the social position of the mother. The analysis contributes to the literature in three ways. First, we estimate the prevalence of BFI and BFD across a wide range of European countries. In contrast to the available research, we will base our rates on data collected from each country that shares a similar design structure and similar measures of BFI and BFD. To date, the Eurobarometer data are the only available source that includes information on both outcomes while adhering to the methodological standards of cross-national data collection. Second, we examine how a number of the well-known individual risk factors of BFI and BFD vary across the different European countries. Finally, by introducing several country-level indicators,

we can examine how policies that specifically target breastfeeding practices, as well as policies that offer parental leave entitlements, relate to both BFI and BFD.

### 1.1. Parental Leave Regulations in Europe

Even though the primary objective of this work is not to provide an exhaustive overview of the European countries' policies—in 2005—it is important to trace out general tendencies and evolutions. A clear overview is provided in leave policies and research by Deven and Moss (2005). There is much variety in constellations of maternity, paternity and parental leave across Europe. Usually, Northern European countries provide generous leave possibilities, with adequate compensation and a broad range of subsidized childcare. Central and Eastern European countries also have quite generous leave possibilities—albeit with less compensation—despite the large socioeconomic changes in the 1990 s. Other European countries are harder to categorize since the variety of leave schemes is vast. However, even in Northern European and Central and Eastern European countries, there are countries deviating from this general leave categorization. When it comes to leave type, maternity leave is usually taken, and until 2005, there has been an increase in uptake of paternal leave. Parental leave is, however, very fragmented and dependent on compensation, job security, and social rights guarantees. All EU states must provide at least three months' leave per parent for childcare purposes. Leave constellations reflect countries' general valuation of care work and responsibilities but are more than mere leave type. It comprises flexibility in working, the organization of services for young children, and incentives to divide care work between parents. Most countries have made an increasing effort to involve fathers in childcare responsibilities by either expanding (obligatory) paternity leave or increasing benefits and incentives for uptake (Deven and Moss 2005).

Fraser (1994) work argued for normative criteria on gender equity in the postindustrial welfare state. Its central thesis is valuing care work and activities as a form of labor work and as a recognized basis for citizenship. Leave regulations that institute entitlements to job-protected time off establish a temporary primacy of early parenthood demands, such as breastfeeding, over workplace demands (Leira 2006). Additionally, they shape concepts of "good" motherhood and organizational work cultures by facilitating the mother's opportunity to breastfeed at home. As Kimbro (2006) explains: 'for working mothers, the decision to return to work and to breastfeed are intricately bound together' (p. 20). By focusing on the extent to which countries value and distribute care work between women and men, the recognition of reproductive differences has the potential to transform traditional gender roles (Ciccia and Verloo 2012).

In the current paper, we examine how variations in BFI and BFD may be related to variations in leave schemes across Europe. Since work and breastfeeding are intricately tied together, a typology should focus on compensatory mechanisms that temporarily alleviate work responsibilities. To explore elements of leave duration and financial compensation, as well paternal leave incentives, we employ Ciccia and Verloo (2012)'s typology, which builds upon Fraser's work on ideal, typical models of the gender division of labor. In contrast to research that focuses on single policy measures, the typology approach considers different policies as interdependent and, therefore, does not allow for compensation effects. For example, if a country offers leaves of short duration, it cannot compensate for this by offering generous financial contributions. Their typology is based on four dimensions: (1) the right to job-protected time off to care for a child; (2) the right to receive monetary compensation during these periods; (3) the distribution of rights between parents; and (4) the presence of incentives for fathers to use leave. In line with Fraser's theoretical model, this typology differentiates countries by the extent to which their leave regulations promote parental childcare in the family and transform traditional gender roles through the regulation of care work. Ciccia and Verloo (2012) distinguish five distinct models of parental leave regulations in Europe, based on their policies from around 2010.

The most common European model—the male breadwinner model—offers parental leaves of three years or more, but these periods are unpaid or paid at low, flat-rate levels—usually below 69% of their wage. There is limited flexibility in taking these leaves, and the scope of incentives for fathers to

take leaves is limited. Countries identified with the male breadwinner model include Austria, Czech Republic, Estonia, France, Hungary, Latvia, Poland, Slovakia, Spain, and the UK.

The second model, the limited caregiver model, is found in several Nordic countries as well as Portugal. Generous compensation, duration of eight months to a year after childbirth, and similar rights for both parents characterize leave regulations in the limited caregiver model. Overall, the Nordic countries with the limited caregiver model encourage parents to return to work within a year after childbirth, but part-time leave makes it possible to spread entitlements over longer periods at a lower level of benefits. Additional periods of full-time leave are available but offered at a lower, flat-rate level. Portugal offers a similar leave system in terms of duration, but the compensation is notably lower.

The third model, the caregiver parity model, offers a leave system similar to the male breadwinner model but also offers mothers generous monetary compensation for leaves. This model is found in three Central Eastern European countries: Bulgaria, Lithuania, and Romania.

Finally, a short duration of leave entitlements typifies the universal breadwinner model, which provides only compulsory maternity leaves intended to protect the health of the mother and the newborn while also securing women's rights to participate in the labor market. Leave compensations do not generally exceed the sickness-leave benefit level since the model assumes that the brevity of entitlements reduces the risk of disincentives to work. Two versions of this model are identified. The unsupported universal breadwinner model offers limited parental leave beyond the compulsory maternity leave, and the additional parental leave is generally not flexible and not well compensated. Countries identified with this model are Belgium, Cyprus, Greece, Ireland, Italy, Malta, and the Netherlands. The supported universal breadwinner model offers leave regulations that are similar but offers mothers higher financial compensations. Countries identified with this model are Denmark, Slovenia, and Switzerland.

As this typology shows, there is substantial variation in how leave schemes are organized, both in terms of duration, financial compensation offered during leave duration, and distribution of parental rights between mothers and fathers. Several studies (Hawkins et al. 2008; Kimbro 2006; Skafida 2012; Guendelman et al. 2009) find that returning to work leads to breastfeeding cessation. Evidence suggests that this return impacts duration more than initiation, the exception being that when mothers return to work within six weeks after childbirth, they may choose not to breastfeed at all (Calnen 2010; Kottwitz et al. 2016). Therefore, we expect that variations in European leave schemes will impact BFD more strongly than BFI, given that the mandatory European minimum duration of protected maternity leave was 14 weeks, with two weeks' compulsory leave before and/or after confinement (Eur-Lex 1992) at the time the data used in this study was collected. However, we note that in some universal breadwinner model countries, a part of that maternity leave may be taken prior to the birth of the child, thereby reducing the maximum length of post-birth maternity leave to periods that the mother perceives as too short for initiating breastfeeding.

Available research also shows that each additional week of maternity leave extends the breastfeeding period by almost half a week (Calnen 2010). Moreover, in Canada, Baker and Milligan (2008) find increasing parental leave entitlements by five weeks and offering a subsequent cash benefit leads to an overall duration increase of a month, with 40 percent of mothers still breastfeeding at six months. These results suggest that longer leaves for mothers will correlate with longer BFD and that this effect will increase when the period is also well-compensated.

Leave entitlements extend to fathers too. The statutory right to extensive parental leave supports breastfeeding duration up to six months by providing fathers the opportunity to share care work responsibilities, leaving a mother with "her time" to breastfeed (Flacking et al. 2007; Evertsson and Boye 2018). During this leave time, there is a fixed period where the mother is highly involved and fully engages with her infant, especially when, subsequently, the father takes paternity or parental leave and is able to set aside his own time to spend with his infant. Given that social support from significant others strongly relates to BFI and BFD (Kools et al. 2006; Ekström et al. 2003; Swanson and Power 2005), the availability of paternity leave creates more opportunities for fathers to support

their spouse during this period. However, uptake of paternity and parental leave by fathers is high only under high-income replacement and extended duration conditions (O'Brien 2009). Consequently, we may expect an increase in both BFI and BFD in countries that offer strong incentives for fathers to use leave.

Taking all of these elements into consideration, we expect that both BFI and BFD will be substantially lower in the unsupported universal breadwinner model, which is characterized by poorly compensated, shorter leave, with few incentives for fathers to take leaves. We expect that initiation will be particularly high in the limited caregiver model, which offers well-compensated, longer leaves. Additionally, the limited caregiver model encourages paternal involvement more, compared to other models, which may also facilitate BFI. However, compared to the caregiver parity model, and to a lesser degree, the male breadwinner model, its leave length generally does not exceed one year, and the vast majority of women return to work after this period (Hegewisch and Gornick 2011). We would therefore expect BFD to be lower in the limited caregiver model than in these two models. Conversely, when flexibility in leave length and uptake—both of which facilitate breastfeeding-family-work balance—are generally more important, we would expect BFD to be higher in the limited caregiver model compared to other models.

## 1.2. Policies that Target Enhancing Breastfeeding Rates

The World Health Organization (WHO) helped design specific policies to support and increase breastfeeding, the most prominent of which are the Baby-friendly hospital initiative (from now on: BFHI) and the international code of marketing of breast-milk substitutes (from now on: the code). The BFHI refers to the certification of hospitals that adhere to the 10 steps for successful breastfeeding (see Table 1). The entire "baby-friendly concept" centers around maternity care as a place where most mothers could be reached (Hofvander 2005).

**Table 1.** Baby-friendly Hospital Initiative, based on the 10 Steps to successful Breastfeeding.

| | **Every Facility Providing Maternity Services and Care for Newborns Should** |
|---|---|
| 1. | Have a written breastfeeding policy that is routinely communicated to all health care staff. |
| 2. | Train all health care staff in skills necessary to implement this policy. |
| 3. | Inform all pregnant women about the benefits and management of breastfeeding. |
| 4. | Help mothers initiate breastfeeding within a half-hour of birth. [a] |
| 5. | Show mothers how to breastfeed and how to maintain lactation even if they should be separated from their infants. |
| 6. | Give newborns infants no food or drink other than breast-milk unless medically indicated. |
| 7. | Practice rooming-in—allow mothers and infants to remain together—24 h a day. |
| 8. | Encourage breastfeeding on demand. |
| 9. | Give no artificial teats or pacifiers (also called dummies and soothers) to breastfeeding infants. |
| 10. | Foster the establishment of breastfeeding support groups and refer mothers to them on discharge from the hospital or clinic. |

[a] In the 2009 revision of the BFHI, this is interpreted as follows: Place babies in skin-to-skin contact with their mothers immediately following birth for at least an hour. Encourage mothers to recognize when their babies are ready to breastfeed and offer help if needed (WHO et al. 2009).

Baby-friendly hospitals (BFH) focus on helping mothers to overcome physiological, emotional, and social barriers during their infant feeding process by, for example, teaching them how to navigate latching problems, encouraging paternal involvement, stimulating rooming-in, and even instructing mothers how to breastfeed when they are separated from their infant (for medical or other reasons). In Switzerland, from 1944 to 2003, BFI levels rose from 22 to 31 weeks and BFD levels from 15 to 17 weeks, in part because of the increase in the number of BFHs during that period (Merten et al. 2005). Similar results have been found in Belarus, where mothers breastfeed more at 12 months under BFHI conditions (Martens 2012). Initiation and duration rates seem to increase *after* official hospital accreditation (Hawkins et al. 2008; Braun et al. 2003). It is particularly important to get mothers with a

precarious social background (i.e., lower maternal education, low family income) into this setting since they tend to have lower breastfeeding rates, especially since mothers who are already motivated to breastfeed will likely choose an environment conducive to their needs (i.e., "baby-friendly").

Similarly, the code is an international health policy program to promote breastfeeding designed by the WHO and adopted by the World Health Assembly (WHA) in 1981 that aims 'to contribute to the provision of safe and adequate nutrition for infants, by the protection and promotion of breastfeeding, and by ensuring the proper use of breast-milk substitutes, when these are necessary, on the basis of adequate information and through appropriate marketing and distribution' (WHO 1981, p. 885). The Code endorses the following principles: the prohibition of advertising and promotion, of giving out samples, and of offering gifts and other inducements, the provision of adequate information and education about young child feeding, the encouragement and promotion of breastfeeding, consumer protection (the infant), and the implementation of the Code. Most studies on the effect of the Code on breastfeeding practices are performed outside the European context and focus on West and Central Africa (Sokol et al. 2008; Aguayo et al. 2003), Pakistan (Salasibew et al. 2008), and Iran (Olang et al. 2009), and only examine the degree to which the Code is implemented and not whether it directly affects BFI or BFD. These studies reveal the occurrence of severe violations of the Code in countries in which it has not been implemented on a national and legal level. The situation in Europe is no different since there aren't many countries where the Code has been translated into national policies (Brady 2012). Worldwide, infant feeding marketing is extremely pervasive and can hinder breastfeeding practices if the Code is not translated on a national policy scale with legitimate legal consequences.

Both policies, the BFHI and the code were specifically designed to enhance and support breastfeeding. The BFHI supports new mothers through difficulties with initiation, empowering them to navigate that breastfeeding challenge. However, since successful long-term breastfeeding is also dependent upon a successful start, a higher prevalence of the BFHI setting in a country can be important for both BFI and BFD but has the potential to influence BFI more. The code focuses more on how alternative infant feeding is marketed and distributed. Depending on the degree of code implementation in a country, mothers are confronted with several formula feeding options either during the later stages of pregnancy or during their maternity ward stay (e.g., by providing free formula samples), or once they return home (e.g., through personalized formula advertising). All aspects of the code relate to stages of the infant feeding decision process, and we expect a stronger adoption of the code will positively affect breastfeeding practices.

## 2. Study Aim and Hypotheses

The aim of the current exploratory study is to examine how countries' social policies shape individual breastfeeding outcomes while also considering the mothers' social position. In order to answer this, we must also address the following sub-questions: what are the BFI and BFD prevalence across Europe, how do well-known individual factors influence BFI and BFD across Europe, and, finally, how do policies and programs specifically targeting breastfeeding as well as policies offering parental leave entitlements relate to BFI and BFD across Europe.

Our hypotheses address one of the unknown elements in research to date, which is how breastfeeding programs/policies and parental leave entitlements relate to BFI and BFD across Europe. With regard to the latter, parental leave entitlements, we focus our first set of hypotheses around the unsupported universal breadwinner model as it is the model offering poorly compensated, shorter leaves, with few incentives for fathers to take leaves. As a starting point, we would expect leave models that offer a combination of extended, well-paid leave, as well as a division of care between parents to score well in terms of breastfeeding.

**Hypothesis 1a (H1a).** *For BFI, all other models (male breadwinner model, limited caregiver model, caregiver parity model and the supported universal breadwinner model) will be associated with higher initiation levels compared to the unsupported universal breadwinner model.*

**Hypothesis 1b (H1b).** *For BFD, only the caregiver parity and supported universal breadwinner model will be associated with longer duration rates compared to the unsupported universal breadwinner model, and we expect no differences between the male breadwinner model and the limited caregiver model, compared to the unsupported universal breadwinner model.*

Conversely, if it is not (well-paid) leave length that is important, and we place emphasis mainly on the flexibility in leave length and uptake for promoting BFD, the following hypothesis is possible.

**Hypothesis 1c (H1c).** *For BFD, the limited caregiver model will be associated with higher duration rates, compared to the unsupported universal breadwinner model.*

Our second set of hypotheses focuses on initiatives specifically designed to support breastfeeding practices, the BFHI and the code. BFHI not only promotes breastfeeding and has several requirements to facilitate easy initiation, but it can also set up a mother for breastfeeding successfully long-term.

**Hypothesis 2 (H2).** *A higher degree of BFHs in the country will be associated with higher initiation and duration chances.*

The code is designed to help protect mothers from marketing strategies of formula feeding; this is regardless of when the mother gave birth. Exposure to marketing would be similar around birth and right thereafter. In the following months, exposure could increase due to several circumstances: return to work, increased outdoor interaction, but also breastfeeding problems. However, if formula feeding marketing is highly regulated, breastfeeding could be protected.

**Hypothesis 3 (H3).** *A higher degree of the code will be associated with a higher initiation and duration chances.*

## 3. Methods

### 3.1. Data

We used Eurobarometer data from 2005, analyzing breastfeeding initiation in 2549 mothers (comparing mothers who have children and breastfeed to mothers who have children but did not breastfeed) and breastfeeding duration in 1929 mothers (analyzing the proportion of mothers who had children, initiated breastfeeding, and breastfed them for several weeks) from 21 countries. Breastfeeding is defined as providing a child with breast-milk, whether directly from the breast or by milk expressed into a bottle. Eurobarometer surveys are implemented on behalf of the European Commission, and the data includes approximately 1000 respondents per country chosen via a multi-stage random probability sampling design. Data were collected from November through December 2005 and is widely accessible for secondary analysis. A complete description of the sampling design can be found in the GESIS Eurobarometer 64.3 report (Papacostas and Magny 2012). To our knowledge, the Eurobarometer 2005 data are the only data that includes comparable information on breastfeeding practices and allows European cross-national comparisons. The initial study sample contained 29,193 respondents. The study excluded mothers over 40 because of the significant time discrepancies between when they breastfed (their last child) or had children and when the data were collected, meaning the largest time-lapse is 22 years (taking 18 years as a marker for first-time motherhood). We added a descriptive per-country table to our results.

The two infant feeding outcomes were breastfeeding initiation and breastfeeding duration. *Initiation* is a dichotomous variable, measuring women who indicated that they had children and breastfed some or all of them (1) and women who had children but did not breastfeed any of them (0). *Duration* is a metric variable measuring the number of months the mother breastfed her last child. It does not consider mothers who bottle-fed from birth on. Since exclusivity was not included in the questionnaire, duration should be seen as the consumption of any type of breast-milk.

We included four contextual measures. The first is *Ciccia and Verloo's typology* (Ciccia and Verloo 2012), which distinguishes the following parental leave types: the limited caregiver model, which includes Portugal, Sweden, and Finland; the unsupported universal breadwinner model (reference category), which includes Belgium, the Netherlands, Italy, Ireland, and Greece; the supported universal breadwinner model, which includes Denmark; the male breadwinner model, which includes France, United Kingdom, Spain, Czech Republic, Estonia, Latvia, Hungary, Poland, and Slovakia; and the caregiver parity model, which includes Lithuania, Bulgaria, and Romania. These data are largely based on data from an EU project, "Quality in Gender Equality Policies" (QUING) (Ciccia and Verloo 2012). For the typology, legislative data from this project was used from 2010 onward. *Baby-friendly hospitals* is a metric variable, measuring the percentage of hospitals in a specific country that received the BFH certificate by the WHO in 2005, according to UNICEF (Hofvander 2005). *The code* is an 8-point scale measuring the amount of government support and legislation that exists for the international code of marketing of breast-milk substitutes. The code aims to legally ban inappropriate advertising of formula feeding. A high score means high legislative involvement (8 = code fully enacted into law; 1 = no governmental/legislative action taken). This scale is based on the UNICEF categorization of the legislative status of the implementation of the code (WHO 2013). At the country level, we also include the *female employment rate* (number of females employed/female population of working age). We retrieved information for the survey period with a time lag of a year (2004) from Eurostat.

In all our models, we controlled for respondent age, education, and employment status as proxies for maternal socioeconomic status and partner status. Age is a proven important precursor to breastfeeding initiation and duration, with older women initiating breastfeeding more and breastfeeding longer (Colodro-Conde et al. 2011). Similarly, education, measured in years of education, is a well-established predictor of infant feeding behavior (Colodro-Conde et al. 2011; Raffle et al. 2011). Current occupational status comprises five categories: housewives (reference group), employed/professional, other white-collar workers, manual workers, and unemployed and non-employed (students, not working due to illness or other). Having a partner can also have a strong supportive or discouraging influence on breastfeeding decisions (Susin et al. 2001). Furthermore, for breastfeeding duration, we controlled for only-child status. Birth order can profoundly influence breastfeeding decisions since the presence of other children may make child feeding less easy (Buckles and Kolka 2014; Isungset et al. 2020).

*3.2. Analysis*

We assessed infant feeding outcomes using multilevel logistic regression analyses for initiation and multilevel linear regression analyses for the duration. Since we analyzed several contextual variables simultaneously in the final models, we performed assumption tests to detect possible multicollinearity issues. However, none were found among the social policies introduced in the models. Covariates are retained based on the aforementioned theoretical assumptions. In line with Bryan and Jenkins (2013) and Stegmueller (2013), we also applied a Bayesian approach to handle the small number of higher-level units. Therefore, we estimated all models with the MLwiN statistical software package using Markov chain Monte Carlo (MCMC) estimation procedures. Regarding the logistic regression analysis, we used y-standardization, as recommended by Mood (2010), to make the odds ratios (ORs) comparable across the nested models. By doing this, we party account for unobserved heterogeneity. All metric-independent variables are grand-mean centered. For each dependent variable, we estimated four models. In the first, we included the individual-level variables in determining whether the differences in breastfeeding initiation and duration can be explained by the composition of the female population (in terms of age, education, employment status, and partner status). In the second, we added Ciccia and Verloo's typology. Thereafter, we included BFHI and the international code stepwise to analyze whether these contextual variables can explain the differences in BFI initiation and duration between parental leave types (Model 3). In the last model (Model 4), we controlled for countries' female employment rate.

## 4. Results

We first consider the descriptive results presented in Figure 2 and Table 2. As Figure 2 shows, there are large differences between countries in both BFI and the average BFD. Additionally, countries with a high percentage of BFI are not necessarily those with the highest average BFD. Denmark, for example, has a high percentage of BFI (94.67%), while its BFD (6.24) is close to the average of the included countries (6.40). In contrast, Spain has a low percentage of BFI (76.67%) compared to the average percentage in other countries (82.56%), but its BFD is close to the overall average (6.41). However, there are some exceptions: Ireland, for example, has a very low percentage of BFI (32.12%) as well as a short average BFD (3.8), while Poland has a relatively high percentage of BFI (91.27%) and a relatively long BFD on average (9.45).

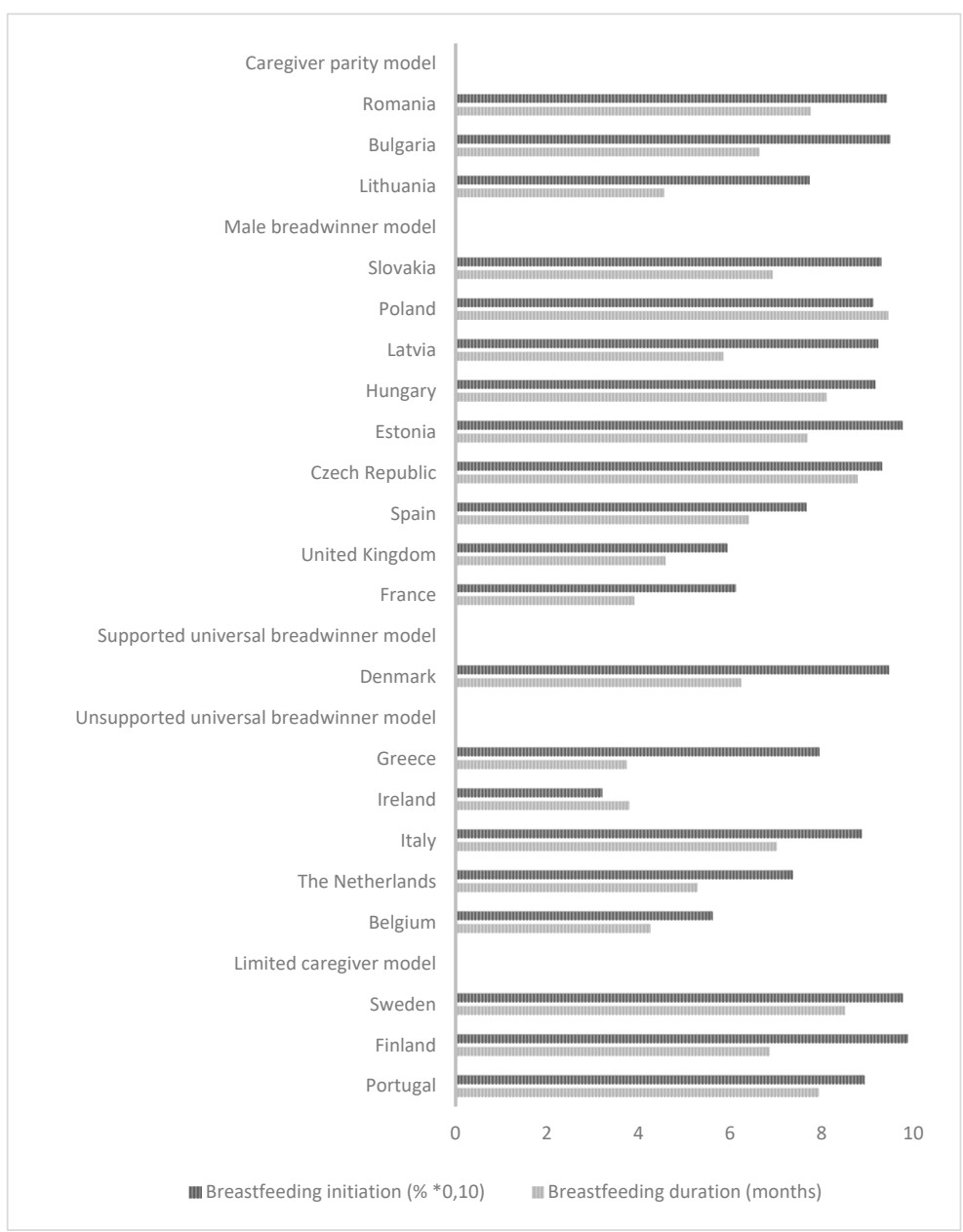

**Figure 2.** Breastfeeding initiation and duration in Europe according to parental leave type, 2005.

**Table 2.** Characteristics of Mothers in Europe, 2005, divided along Parental leave Types (means, SD and percentages).

| | Maternal Level | | Country Level | | |
| --- | --- | --- | --- | --- | --- |
| **Parental Leave Type** | **Breastfeeding Initiation** | **Breastfeeding Duration** | **BFHs** | **The code** | **Female Employment Rate** |
| | % | Average Number of Months (SD) | % (SD) | Index Score (SD) | % |
| **Limited caregiver model** | **94.95** | **7.77 (6.16)** | **34.86 (42.52)** | **6.69 (0.46)** | **70.06** |
| Portugal | 89.32 | 7.93 (6.80) | 0 | 7 | 65.66 |
| Finland | 98.85 | 6.86 (4.53) | 14 | 7 | 69.56 |
| Sweden | 97.70 | 8.51 (6.77) | 97 | 6 | 75.78 |
| **Unsupported universal breadwinner model** | **67.58** | **5.30 (4.77)** | **1.01 (2.46)** | **6.69 (0.92)** | **55.36** |
| Belgium | 56.19 | 4.26 (3.81) | 0 | 7 | 56.10 |
| The Netherlands | 73.75 | 5.29 (6.11) | 7 | 7 | 66.08 |
| Italy | 88.79 | 7.02 (5.29) | 0 | 7 | 45.00 |
| Ireland | 32.12 | 3.80 (3.17) | 0 | 7 | 60.78 |
| Greece | 79.49 | 3.74 (3.19) | 0 | 7 | 46.92 |
| **Supported universal breadwinner model** | **94.67** | **6.24 (3.63)** | **18.00 (0.00)** | **7 (0.00)** | **73.02** |
| Denmark | 94.67 | 6.24 (3.63) | 18 | 7 | 73.02 |
| **Male breadwinner model** | **84.18** | **6.96 (6.57)** | **5.39 (4.80)** | **4.73 (1.95)** | **59.94** |
| France | 61.29 | 3.91 (3.56) | 0 | 7 | 63.32 |
| United Kingdom | 59.39 | 4.59 (4.07) | 14 | 7 | 67.46 |
| Spain | 76.67 | 6.41 (10.94) | 0 | 7 | 48.28 |
| Czech Republic | 93.15 | 8.78 (7.05) | 9 | 3 | 61.78 |
| Estonia | 97.60 | 7.69 (6.69) | 0 | 5 | 64.78 |
| Hungary | 91.74 | 8.10 (6.59) | 8 | 5 | 54.62 |
| Latvia | 92.31 | 5.85 (5.14) | 0 | 3 | 61.78 |
| Poland | 91.27 | 9.45 (8.15) | 5 | 3 | 52.20 |
| Slovakia | 92.99 | 6.93 (5.14) | 7 | 2 | 57.40 |
| **Caregiver parity model** | **88.86** | **6.44 (5.18)** | **2.02 (2.17)** | **2.00 (0.00)** | **58.82** |
| Lithuania | 77.31 | 4.56 (4.47) | 0 | 2 | 64.26 |
| Bulgaria | 94.96 | 6.64 (5.57) | 1 | 2 | 52.90 |
| Romania | 94.21 | 7.76 (4.91) | 5 | 2 | 59.30 |

We find a large amount of cross-national variation for BFI, with a percentage difference of 66.73 between countries, with the lowest percentage in Ireland (32.12%), followed by Belgium (56.19%), and the highest in Finland (98.85%), followed by Sweden (97.70%). Table 2 shows an average BFD range from 3.75 in Greece to 9.45 in Poland. The lowest prevalence rates in BFI and the shortest average BFD are observed in a number of unsupported universal breadwinner model regimes. For example, Belgium, the Netherlands, Ireland, and Greece report low prevalence rates in BFI and a short average BFD. The highest prevalence rates for BFI and long BFDs are found in the limited caregiver model, particularly in the Scandinavian countries Finland and Sweden, as well as in the supported universal breadwinner model (Denmark), and two countries in the caregiver parity model, Bulgaria and Romania.

Based on this descriptive table, we conclude that the international code has a stable value of seven (highest degree of implementation) across countries with the limited caregiver model (excl. Portugal with a degree of six), the unsupported universal breadwinner model, and the supported universal breadwinner model. France, the United Kingdom, and Spain (countries with a male breadwinner model) also have a degree of seven. All countries with the caregiver parity model, as well as Slovakia, have the lowest degree of implementation of the code, 2. Furthermore, we notice that the BFH score is high in Sweden, moderate in Finland, Denmark and the UK, and low in the South European countries' regimes. There is no clear pattern according to parental leave types. Furthermore, we observe that the highest rates of female employment occur in countries with a limited caregiver model or supported universal breadwinner model.

In addition, to mean scores and percentages, we report the variance decomposition for our higher-level variables, revealing the following points. First, from the null-model variance decomposition, we notice that for breastfeeding initiation, a large part of this infant feeding choice is determined by country-level differences (29.6%), meaning that for more than one in four mothers,

the feeding decision is influenced by country-specific characteristics. For breastfeeding duration, the null-model variance percentage is lower yet still significant: 7.7% of the variance in breastfeeding duration is situated at the country level.

Subsequently, we estimate random intercept models for all 21 countries in the sample. The odds ratios (OR) for BFI presented in Table 3 give the odds of women initiating breastfeeding compared to women, not breastfeeding. An OR higher than one would indicate higher odds of BFI; conversely, an OR lower than one would significantly lower odds of BFI.

**Table 3.** Determinants of Breastfeeding Initiation in Europe, 2005 (N = 2549).

| Variables | Model 1 | | Model 2 | | Model 3 | | Model 4 | |
|---|---|---|---|---|---|---|---|---|
| **Fixed Part** | **OR** | **CI** | **OR** | **CI** | **OR** | **CI** | **OR** | **CI** |
| **Individual level** | | | | | | | | |
| Cons[1] | 2.470 *** | (1.512–1.805) | 1.328 | (0.721–1.646) | 1.758 * | (1.079–2.940) | 1.559 | (0.916–1.752) |
| Maternal age | 1.012 | (0.999–1.024) | 1.012 | (0.999–0.025) | 1.013 | (1.000–2.718) | 1.013 | (0.999–1.025) |
| Maternal education | 1.034 ** | (1.012–1.056) | 1.033 ** | (1.010–1.054) | 1.033 ** | (1.010–2.745) | 1.033 ** | (1.010–1.054) |
| Partner (ref. no partner) | 1.160 ° | (0.996–1.273) | 1.152 | (0.987–1.273) | 1.147 | (0.986–2.679) | 1.150 | (0.986–1.271) |
| **Employment status** | | | | | | | | |
| Housewife (ref.) Self-employed/professional | 0.984 | (0.654–1.245) | 0.980 | (0.627–1.234) | 0.972 | (0.615–1.231) | 0.983 | (0.634–1.240) |
| Other white-collar worker | 0.959 | (0.751–1.121) | 0.948 | (0.745–1.104) | 0.945 | (0.743–1.099) | 0.946 | (0.751–1.097) |
| Manual worker | 0.795 * | (0.416–0.959) | 0.781 * | (0.375–0.943) | 0.773 * | (0.360–0.929) | 0.771 * | (0.359–0.924) |
| Unemployed | 0.717 ** | (0.203–0.849) | 0.712 ** | (0.199–0.846) | 0.704 ** | (0.192–0.825) | 0.704 ** | (0.191–0.824) |
| Non-employed | 0.774 | (0.010–1.212) | 0.766 | (0.000–1.177) | 0.760 | (0.000–1.171) | 0.750 | (0.000–1.151) |
| **Country level** | | | | | | | | |
| Parental leave type | | | | | | | | |
| Unsupported universal breadwinner model (ref.) | | | | | | | | |
| Limited caregiver model | | | 2.676 ** | (1.393–2.035) | 2.464 ** | (1.291–3.637) | 2.956 ** | (1.458–2.081) |
| Supported universal breadwinner model | | | 1.933 | (0.653–2.057) | 1.953 | (0.748–2.112) | 2.504 * | (1.145–2.119) |
| Male breadwinner model | | | 1.737 * | (1.067–1.796) | 1.261 | (0.512–1.669) | 1.407 | (0.728–1.714) |
| Caregiver parity model | | | 1.916 * | (1.045–1.915) | 0.956 | (0.002–1.765) | 1.081 | (0.001–1.764) |
| BFHs | | | | | 1.001 | (0.984–2.674) | 1.005 | (0.988–1.022) |
| International code | | | | | 0.830 | (0.531–1.298) | 0.842 * | (0.567–0.986) |
| Female employment rate | | | | | | | 0.971 | (0.927–1.010) |
| Variance | | | | | | | | |
| Country level | 1.664 | −0.615 | 1.227 | 0.529 | 0.979 | 0471 | 0.862 | 0.457 |
| DIC | | | 1,941,636 | | 1,942,453 | | 1,943,047 | |

Significance: ° *p* < 0.10; * *p* < 0.05; ** *p* < 0.01; *** *p* < 0.001. Deviance Information Criterion (DIC).

We include individual variables in the first model of Table 3. We find significant relationships only between BFI and women's educational level and employment status/type. Higher educated women are slightly more likely to initiate breastfeeding (OR = 1.034), and manual workers (OR = 0.795) and unemployed women (OR = 0.717) are slightly less likely to initiate breastfeeding compared to housewives. Having a partner is marginally significantly related to breastfeeding initiation (at the 10% level): Women with a partner are somewhat more likely to initiate breastfeeding (OR = 1.060).

Next (Model 2), we add parental leave types. As expected, in countries with an unsupported UBM, women are least likely to initiate breastfeeding. In limited caregiver model countries, the likelihood to initiate breastfeeding is significantly higher when compared to unsupported universal breadwinner model countries (OR = 2.676) and when compared to the caregiver parity model (OR = 1.916) and the male breadwinner model (1.737) countries. BFI in the supported universal breadwinner model does not differ significantly from BFI in unsupported universal breadwinner model countries.

The breastfeeding-specific social policies show an association with BFI in the opposite direction than what we expected (Model 3). We found that countries with a higher degree of international code implementation show lower levels of BFI (OR = 0.830) than countries with a lower degree or

---

[1]    Intercept.

no degree of implementation. The BFHs show no significant association with BFI. We also find that the male breadwinner model and the caregiver parity model no longer differ significantly in BFI from the unsupported universal breadwinner model when considering these breastfeeding policies. This may indicate that the differences in BFI between these models can be (partly) ascribed to a country's degree of implementation of the international code, as caregiver parity model and male breadwinner model countries have a lower degree of implementation. In our last model, we also control for countries' female employment rates. Although there is no significant relationship between the female employment rate and BFI, by controlling for it, the difference in BFI between supported and unsupported universal breadwinner models becomes significant. In supported universal breadwinner model countries (Denmark), mothers are significantly more likely to initiate breastfeeding compared to mothers in unsupported universal breadwinner model countries; this may be associated with the female employment rate being higher in Denmark than in unsupported universal breadwinner model countries (OR = 2.504). In addition, the positive relationship between the limited caregiver model and BFI (compared to the unsupported universal breadwinner model) becomes stronger (OR = 2.956).

When looking at the coefficients presented in Table 4, which focus on BFD, we see in Model 1 that only women's employment status/type is significantly associated with BFD. White-collar women display a significantly shorter BFD than housewives (b = −1.011 [0.336]). In Model 2, we observe largely the same pattern as we do for BFI. BFD is significantly longer in the limited caregiver model (b = 3.025 [1.278]) and the male breadwinner model (b = 1.957 [0.956]) countries compared to countries with an unsupported universal breadwinner model. However, this is not completely in line with our expectations. Moreover, the caregiver parity model did not significantly differ from the unsupported universal breadwinner model in terms of BFD.

**Table 4.** Determinants of Breastfeeding duration in Europe, 2005 (N = 1929).

| Variables | Model 1 | | Model 2 | | Model 3 | | Model 4 | |
|---|---|---|---|---|---|---|---|---|
| **Fixed Part** | b | SE | b | SE | b | SE | b | SE |
| **Individual level** | | | | | | | | |
| Cons$^2$ | 6.629 *** | 0.566 | 5.117 *** | 0.989 | 6.466 *** | 0.965 | 5.980 *** | 1.028 |
| Maternal age | 0.028 | 0.027 | 0.029 | 0.027 | 0.031 | 0.027 | 0.031 | 0.027 |
| Maternal education | 0.021 | 0.039 | 0.017 | 0.038 | 0.011 | 0.039 | 0.015 | 0.038 |
| Partner (ref. no partner) | 0.459 | 0.361 | 0.459 | 0.357 | 0.446 | 0.351 | 0.472 | 0.355 |
| Employment status | | | | | | | | |
| Housewife (ref.) Self-employed/professional | 0.108 | 0.557 | 0.112 | 0.558 | 0.096 | 0.550 | 0.087 | 0.549 |
| Other white-collar worker | −1.011 ** | 0.336 | −1.048 ** | 0.331 | −1.082 ** | 0.331 | −1.059 ** | 0.332 |
| Manual worker | −0.185 | 0.451 | −0.236 | 0.449 | −0.282 | 0.449 | −0.277 | 0.451 |
| Unemployed | −0.868 | 0.500 | −0.897 ° | 0.488 | −0.922 ° | 0.490 | −0.951 ° | 0.491 |
| Non-employed | −1.780 | 1.070 | −1.810 | 1.083 | −1.845 | 1.071 | −1.843 | 1.090 |
| Only child (ref. no) | −0.267 | 0.324 | −0.259 | 0.321 | −0.272 | 0.318 | −0.259 | 0.318 |
| **Country level** | | | | | | | | |
| Parental leave type Unsupported universal breadwinner model (ref.) | | | | | | | | |
| Limited caregiver model | | | 3.025 * | 1.278 | 2.459 | 1.279 | 3.284 * | 1.430 |
| Supported universal breadwinner | | | 1.597 | 1.919 | 1.466 | 1.662 | 2.576 | 1869.000 |
| Male breadwinner model | | | 1.957 * | 0.956 | 0.510 | 0.957 | 0.909 | 0.992 |
| Caregiver parity model | | | 1.677 | 1.265 | −1.587 | 1.586 | −1.126 | 1603.000 |
| BFHs | | | | | 0.010 | 0.019 | 0.017 | 0.020 |
| International code | | | | | −0.656 ** | 0.236 | −0.620 ** | 0.234 |
| Female employment rate | | | | | | | −0.067 | 0.055 |
| **Variance** | | | | | | | | |
| Country level | 3.027 | 1.196 | 2.619 | 1.162 | 1.740 | 0.872 | 1.669 | 0.879 |
| Individual level | 29.616 | 0.978 | 29.611 | 0.965 | 29.600 | 0.964 | 29.613 | 0.966 |
| ICC | 0.093 | | 0.081 | | 0.056 | | 0.053 | |
| DIC | | | 12,039.322 | | 12,037.980 | | 12,038.362 | |

Significance: ° *p* < 0.10; * *p* < 0.05; ** *p* < 0.01; *** *p* < 0.001. Intra-Class Coefficient (ICC), Deviance Information Criterion (DIC).

In the subsequent model, we find the same negative effect for the international code: countries that have implemented the international code to a greater degree display shorter BFD, while countries that have implemented it to a lesser degree or have not implemented it at all display longer BFD (−0.656 [0.256]). Additionally, the relationship between parental leave type and BFD is no longer significant. By adding the breastfeeding-specific policies, the country-level variation decreases from 8.1% to 5.8%.

In the last model, we add the female employment rate, but like BFI, it has no significant effect on BFD. However, the positive relationship between the limited caregiver model (compared to the unsupported universal breadwinner model) and BFD becomes significant again after controlling for female employment rate (b = 3.284 [1.430]).

## 5. Discussion

Our main research question—how countries' social policies shape individual breastfeeding outcomes while considering the mother's social position—addresses three other questions: on prevalence across Europe, well-known individual characteristics and policy/program influences. First, on the prevalence of BFI and BFD in Europe, our study finds considerable cross-national variation. The highest BFI rates are found in Finland and Sweden, the lowest in Ireland and Belgium. Duration of breastfeeding is substantially longer in countries such as Poland and the Czech Republic, while countries such as France and Greece have the shortest duration rates in Europe. Additionally, some countries with low BFI rates nonetheless have duration rates similar to other European countries. These findings suggest that low initiation does not automatically translate to low duration. Several of our findings also support the relationship between cross-national variation and both the composition of the maternal population within the different European countries and the setup of the broader institutional context.

Second, our study confirms the social gradient in breastfeeding initiation but not in breastfeeding duration. We find that higher educated women are more likely to initiate breastfeeding compared to lower educated women but not more likely to continue breastfeeding for longer periods of time. The positive effect of higher education on breastfeeding initiation has been linked to both higher levels of health literacy and higher levels of self-efficacy and to greater social support for higher educated women (Scott and Binns 1999; Chezem et al. 2003; Riva et al. 1999; Cernadas et al. 2003; Mirowsky and Ross 2005). Higher educated women are also more likely to endorse an intensive motherhood ideology (Fox 2006), which may explain their higher likelihood to initiate breastfeeding within a context that promotes breastfeeding as a desirable motherhood practice. Other research, however, shows that long-duration breastfeeding has significant labor market pay-offs, which may be pronounced in higher educated women, especially (Rippeyoung and Noonan 2012). Several authors also theorize that an educational *degree*, with its associated skills and competencies, rather than cumulative years of education, affects general health (Lynch 2006; Mirowsky and Ross 2005). Unfortunately, our data included information on only the latter measure of education. We also find that unemployed women and women in manual labor are less likely to initiate breastfeeding than women who are homemakers, while women in other white-collar occupations breastfeed for a shorter time than homemakers. Our research does not support a time-availability argument, where working fewer or no hours in paid employment indicates higher breastfeeding initiation rates, since we find that unemployed mothers initiate less than housewives, while employed women are as likely to initiate breastfeeding as housewives. Flexibility could be an issue since manual workers also initiate less, and their job structure is typically more rigid. Also, the interaction between work status and leave uptake is relevant because shorter leaves lead to discontinuation, regardless of part-time employment versus full-time work status (Mandal et al. 2010). Part-time employment is not always an option

---

2   Intercept.

for mothers or employers, leading to choosing between other options (i.e., breastfeeding breaks or flextime) to facilitate breastfeeding. Further research that considers time management, flexibility, and job structure are needed to explore why some jobs within certain countries protect breastfeeding and others do not.

Third, we find substantial differences between breastfeeding practices across the different leave regulation models. For BFI, the highest rates were found in the limited caregiver model, partially confirming H1a. This model offers longer, well-compensated leaves and provides uptake flexibility (Ciccia and Verloo 2012), which may facilitate BFI. It was unexpected that the only other model with higher BFI rates was the supported universal breadwinner model, which is typically characterized by shorter, underpaid leaves. Unfortunately, while there are several European countries classified as having a supported universal breadwinner model, the current analysis covers only one country, Denmark. In 2005, Denmark had 4 pre-birth and 14 postnatal weeks of maternity leave and parental leave of up to 46 weeks but paid at a flat-rate level for only one parent (OECD 2017). Thus, in Denmark, mothers have a fixed maximum number of weeks to take pre-birth leave, leaving a significant post-birth maternity period with ample room for mothers to initiate breastfeeding, especially when this period is combined with extended parental leave, even when compensated at a flat-level rate. Moreover, Denmark's female employment rate ranks among the highest in Europe. This shows that high female employment does not always hinder breastfeeding initiation.

For BFD, only the limited caregiver model countries have higher duration rates compared to the unsupported universal breadwinner model, endorsing H1c over H1b. Clearly, flexibility in well-paid leave uptake and division of leave between parents is more important than mere leave length. Extending paternal or parental leave, with attention to changing gender norms and traditional gender stratification, creates a situation where care can be divided amongst parents, providing more opportunity for mothers to breastfeed for several months (Flacking et al. 2007). The incentives for fathers to take leave are greater not only when financial compensation is high (50% or higher), but also when it is possible to take *extended* leave (O'Brien 2009). For most of our countries, leave is usually specific to the mother and/or father and not dependent on the other, taking parental leave, which would benefit BFD. The exceptions are Bulgaria, Portugal, Spain, the Czech Republic, and Poland, where part of maternity leave is transferrable. Cultural norms, however, place the primary responsibility for care with mothers (Ciccia and Verloo 2012), which makes transferring maternity leave time to the father unlikely. Given that the highest average European duration rate is just under 10 months, mothers may benefit from other types of governmental support, besides parental leave length schemes, to facilitate prolonged breastfeeding. Our results indicate that most differences between the parental leave models disappear when adjusted by cross-national differences in breastfeeding-targeted policies and in female employment rates.

For both BFI and BFD, there is little difference between the unsupported universal breadwinner model, the male breadwinner model, and the caregiver parity model, and little difference found in BFD in the supported universal breadwinner model. Although our typology considers job-protected time off from work, the amount of paid leave, paternal involvement, and distribution of rights, what remains unknown is the amount of leave taken by parents. This is in line with Mandal et al. (2010), where controlling for actual leave taken attenuates general maternity leave effects. European variation in mandatory versus non-mandatory maternity, paternity, and parental leave is considerable (Van Belle 2016). Individual entitlement is relevant specifically for parental leave since it encourages fathers or partners to be more involved in care work and supporting their spouse (Flacking et al. 2007; O'Brien 2009). In their work, Grunow and Evertsson (2016) describe how couples transition into parenthood and the broad variation existing across European countries. Individual strategies are inextricably linked to the broader policy design and possibilities offered by the nation. For breastfeeding specifically, balancing breastfeeding with the return to work is an important precursor, with flexible measures (i.e., possibility of transitioning to part-time employment or taking breastfeeding breaks) and available childcare options (i.e., privatized or public, and their funding) as determinants of

breastfeeding difficulties. Grunow and Evertsson (2016) further describe how breastfeeding is often "expected" and the "normal" choice to make, which indicates how individual motivation is also strongly connected to the societal expectation to breastfeed and preexisting norms.

The breastfeeding-specific policies also play a distinctive role in explaining cross-national variation in BFI and BFD. We find that BFHs do not appear to be positively associated with initiation or duration rates, meaning H2 is not confirmed. Introducing and implementing the BFHI has been a difficult task in many countries (Hofvander 2005). As mentioned above, in the original "baby-friendly-concept," the focus was almost entirely on maternity centers as places where most mothers could be reached (Hofvander 2005). However, Europe is typified by a variety of both health care facilities and models of maternity care (Christiaens 2008). Cross-national differences may often be missed because BFHs obtain different results in different countries, especially because not only the practices associated with BFHs but also the breastfeeding-friendly label affects breastfeeding rates (Labbok 2007). Moreover, our analyses measure policies at one moment in time, whereas their evolution might be more important (Merten et al. 2005). Whether rapid or slow, a change in policy development could indicate differences in infant feeding attitudes, norms, and values, and may, furthermore, influence BFI and BFD, but only when these changes are well-established. However, the practices associated with the BFHI concept may already be part of a country's tradition, making accreditation unnecessary to produce the desired breastfeeding outcomes (e.g., Finland and Sweden have similar high breastfeeding rates, but only 14% of BFHs in Finland are accredited compared to Sweden's 97%). BFHs can thus play a supportive role for countries where breastfeeding is or has not been established individual behavior. In this sense, it is also important to examine other gendered macrolevel predictors. For example, a recent study by Hoxha et al. (2020) and colleagues demonstrated a lower c-section rate for countries with a higher representation of female physicians. Cesarean sections form a persistent barrier to breastfeeding initiation, especially regarding elapsed time between birth and first time breastfeeding which is step four in the BFHI program (Rowe-Murray and Fisher 2002).

Contrary to expectations and existing literature, we find that the International code is negatively associated with BFI and BFD, meaning H3 is also not confirmed. It may be that the effects of these policies are country-specific, dependent on the practical implications of a country's legislative framework (e.g., the penalization for violations or promotional efforts, or the way in which the dialogue with the food industry is constructed), instead of reflecting cross-national differences. Both the implementation and monitoring of the International code remain important if it is to positively affect breastfeeding rates. Because the code scores low across caregiver parity model countries but is high or constant in the unsupported and supported universal breadwinner models and the limited caregiver model, the code may affect the caregiver parity model more strongly than the other types. The code scores high in regime types with high breastfeeding rates, as well as regime types with low breastfeeding rates. The male breadwinner model is the only model where the code shows significant between-country variation. Moreover, we do not know to what extent the International code is implemented in specific health facilities and among health care workers in Europe. For example, a meta-analysis of the effect of commercial discharge packs (artificial milk samples, bottles, teats) reveals them to be detrimental to breastfeeding rates (Perez-Escamilla et al. 1994). Since some model types score well on BFI and BFD as well as the code, it could indicate a country does not depend on the code to positively impact breastfeeding patterns. Alternatively, the index the code is based on could be insufficient in capturing marketing differences between countries, and maternity care centers, leading to our confounding results. Therefore, these breastfeeding-specific policies should be examined further from a cross-national perspective, especially since previous research has found them to be important in various cross-sectional studies (Merten et al. 2005; Martens 2012).

*Limitations*

The Eurobarometer presents a unique opportunity to study infant feeding processes from a comparative perspective; however, it has certain limits that may affect the comparability of multi-country

studies, such as translations, cultural interpretations, conduct, and methodological differences. In our study, one limitation is our use of 14-year-old data. Our objective was, however, to gain insight into how social policies help shape breastfeeding behavior, not to analyze current European breastfeeding policies. Even though our data are older, it still provides new insight into cross-national infant feeding differences that did not exist before, upon which future research can build. The current leave policy situation in Europe is mostly an expansion of leave time for mothers, fathers, and parents, and does include some incentives for a father to take up leave (Koslowski et al. 2019), but could focus more on extended flexibility measures to ensure an adequate work-family balance between parents. We would expect increased leave time to benefit breastfeeding rates in countries where flexibility and an equal division of care, reflected in a change in the role of the mother as a traditional caregiver, is already in place. Eurobarometer data are further limited due to its smaller country samples. Ideally, a larger variety of mothers per country would provide a more accurate representation of breastfeeding practices within that country but also a stronger basis for cross-national comparisons. Another limitation is the use of individual sociodemographic information from 2005 as a proxy for the sociodemographic information at the time of childbirth. This effect was attenuated by constricting the respondent's age interval (max. 40 years) to minimize time discrepancies. The exclusion of mothers over 40 is not ideal. However, this choice was not random but based on age at last childbirth (marking the end of female fertility) from a demographic perspective, which ends at approximately 40–41 years of age in natural fertility populations (ESHRE Capri Workshop Group 2010). By filtering out data and selecting only more static sociodemographic information, we gain indicators that are more reliable and proven to be important determinants in infant feeding (Singh et al. 2007; Colodro-Conde et al. 2011). In addition, the measure for breastfeeding duration may obscure important between-country differences in the composition of infant feeding determinants among European mothers. We do not know how the exclusivity of breastfeeding was measured and interpreted among mothers. Finally, the cross-sectional nature of our data does not allow any causal interpretations of the results. Regarding breastfeeding duration, there is the possibility of an overrepresentation of breastfeeding mothers in our population since they only were examined at one point in time. These data could represent a specific subset of mothers whose participation in the project did not hinder family and/or work obligations, leading to breastfeeding mothers who had time to breastfeed.

## 6. Conclusions

Current research on breastfeeding from a comparative perspective is scarce, particularly within high-income countries. The results of our main research question—how countries' social policies and mothers' individual social positions shape breastfeeding outcomes—reveal both the complexity of and the need for more cross-national research. Because the "breast is best" message is widespread across Europe (Lee 2008; Saguy 2011), it has become a moral dilemma for some mothers even if they are able to breastfeed and are therefore considered "good mothers" (Wall 2001). As Katz-Rothman (2000) states, "the breast is particularly problematic—deeply sexualized, racialized, and contested." This leads to the ongoing discussion on who makes feeding decisions: between husbands and wives, between family and friends' influence, among medical professionals, and in turn, between countries that create policies and procedures directly affecting a mother's opportunity to breastfeed.

When comparing countries on breastfeeding outcomes, Portugal, Sweden, and Finland (as well as Denmark) score high compared to the other European countries, these countries highlight flexibility in well-paid maternal and paternal leaves and an equitable division of parental leave between parents, which essentially refers to the different underlying gender ideologies and corresponding gender norms on the appropriation of care work. However, future research needs to consider the interaction with individual entitlements. Employment status and occupation, flexibility in the workplace, and general financial resources (Rippeyoung and Noonan 2012), along with the general societal acceptance of supporting care work (Fraser 1994; Wall 2001), shape breastfeeding outcomes in conjunction with general leave entitlements. This relates to the amount of agency a mother *feels* she has when

breastfeeding (Mackendrick 2014) and to the constraints social structures put on a mother (Williams 2001). The influence structure exerts over individual behavior is partially determined by political motivations and the current social problems societies face (Wennemo 1992). This becomes especially poignant in light of the current worldwide pandemic, COVID-19. Since many people are strongly encouraged or forced to work from home due to nationwide lockdowns, boundaries between work and family become blurry or start disappearing altogether. Compensatory mechanisms are needed to alleviate many pressures faced by young parents, especially for women who became mothers during this pandemic. Nations are in a unique position to contribute to a valuable discussion on valuing women's (re)productive work, and achieving an adequate work–life balance across the board.

In sum, we provide a range of results that are applicable to all policies considered in this study; our study is, therefore, able to consider nuanced differences in infant feeding processes. Some findings are somewhat surprising and clearly require further analysis. Current analyses do not fully capture motherhood ideology in feeding decisions. Breastfeeding is linked with the concrete conceptualization of motherhood (Lee 2008), where specific discourses, such as breastfeeding as natural in society or the female body as a mere producing machine, directly enhance or impair breastfeeding practices, respectively (Wall 2001; Larsen et al. 2008). Breastfeeding may fulfill an engrained traditional gender role pattern or satisfy a progressive health behavior related to good mothering (Kronborg and Vaeth 2004); different countries employ a variety of strategies to reward, outsource, or oversee this type of care work. This does suggest that the infant-feeding tradition is deeply rooted in countries and extends beyond legislation and that it refers to the (de)construction of gendered parenting models and parenting cultures as well as the politics of reproduction itself (Latimer and Thomas 2017; Sutton 2008). Perhaps by examining specific social policies separately, or by focusing on the interplay between the different constructions of maternity, paternity, and parental leave and individual motivation, we could better understand these cross-national differences. Nonetheless, our main results suggest the existence of an important contextual and social policy dimension to the issue of infant feeding. It seems that changing gender ideology and gender norms among parents, reflected in policy about paid maternity and parental leave length, uptake flexibility, and paternal involvement, all of which influence breastfeeding rates, should put the issue of infant feeding on governmental policy agendas across European countries.

**Author Contributions:** Conceptualization, K.V.; methodology, V.B.; software, V.B.; formal analysis, V.B.; validation, V.B. and K.V.; investigation, K.V. and S.V.d.V.; data curation, K.V.; writing—original draft preparation, K.V., S.V.d.V., V.B. and B.V.d.P.; writing—review and editing, K.V.; visualization, K.V. and V.B.; supervision, B.V.d.P. and S.V.d.V.; project administration, S.V.d.V. and K.V. All authors have read and agreed to the published version of the manuscript.

**Funding:** This research received no external funding.

**Conflicts of Interest:** The authors declare no conflict of interest.

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
