# Peer review of "Motherhood in Europe: An Examination of Parental Leave Regulations and Breastfeeding Policy Influences on Breastfeeding Initiation and Duration"

_socsci, doi:10.3390/socsci9120222_

Round 1

Reviewer 1 Report

This is an important study that reviews the literature very comprehensively, and carries out a good analysis of the situation using their models of breastfeeding presented. In the midst of the Covid-19 pandemic it is unclear what impact your study might help on policy in the near future.

Line 102 talks about national C-section rates. A recent study demonstrates lower rates with more female doctors. Hoxha, I., F. Sadiku, A. Lama, G. Bunjaku, R. Agahi, J. Statovci and I. Bajraktari (2020). "Cesarean Delivery and Gender of Delivering Physicians: A Systematic Review and Meta-analysis." Obstetrics & Gynecology. It would be of interest to add female doctor proportions in each nation to the analysis if these data are available.

Author Response

Response to reviewer 1

We appreciate the time you have taken to read our work. Below, we explain how your remarks were addressed throughout the manuscript. All review changes are highlighted throughout the manuscript.

This is an important study that reviews the literature very comprehensively, and carries out a good analysis of the situation using their models of breastfeeding presented. In the midst of the Covid-19 pandemic it is unclear what impact your study might help on policy in the near future.

  • Thank you for suggesting this. It is especially relevant to address the pandemic and how policy has an important role to play in order to support parents. We have added an extra paragraph in the conclusion section to address this.

Line 102 talks about national C-section rates. A recent study demonstrates lower rates with more female doctors. Hoxha, I., F. Sadiku, A. Lama, G. Bunjaku, R. Agahi, J. Statovci and I. Bajraktari (2020). "Cesarean Delivery and Gender of Delivering Physicians: A Systematic Review and Meta-analysis." Obstetrics & Gynecology. It would be of interest to add female doctor proportions in each nation to the analysis if these data are available.

  • As you point out, we expect there are other macro-level determinants when it comes to breastfeeding behavior. Unfortunately, we are unable to add these to our models. To address this issue, we have added this suggestion to our discussion on baby-friendly hospitals so future research can focus on other gendered macro-level determinants.

Thank you for taking the time to read our revised manuscript.

Sincerely,

The authors

Reviewer 2 Report

The manuscript addresses an important and so far not sufficiently explored question, and does so in a systematic, thorough and methodologically sound manner. It uses a multilevel design that makes it possible to carry out an analysis including 21 European countries and a variety of policy regimes. Its findings are rich and conclusions fill several gaps in the existing literature. For these reasons, I consider the article should be published. I would nonetheless recommend considering some minor questions:

-Although the manuscript does make an important contribution to earlier research, greater emphasis could be made on both its scientifical and social relevance in the background section, so that it is quickly conveyed to the reader (for instance, one strength of the article that is not sufficiently underscored is that it actually contributes to different streams of research/disciplines/areas of study: the literature on breastfeeding/public health, earlier research on parental leave and policy effects, the socioeconomic determinants of health-related behaviors, research on parenthood behavior, etc.). 

-The manuscript considers the relevant literature on the topic. Nevertheless, its theoretical framing (leading to the hypotheses formulated) could be strengthened. The exact expected theoretical mechanisms leading to each of the hypotheses should be spelled out more clearly. It could be useful to shorten the background section somewhat and instead expand the "study aim and hypotheses section" with more explicit explanations of the mechanisms expected to be at play in the different types of policy regimes. 

-It should be clarified in the background section that the classification of countries in the typology used refers to their policies in 2005. It should be clarified how much time had elapsed, on average, between the date of birth and the date of the survey.

-Parental leave in the male breadwinner model is claimed to be unpaid or paid at low, flat-rate levels, usually below 60% of the wage. Nevertheless, this is not always the case - Spain f.i. was an exception in 2005, with 100% paid maternity leave (yet with a maximum ceiling) during 16 weeks; even though longer parental leave was unpaid). It could be useful to check whether there were any other such exceptions 2005 (perhaps Moss & O'Brien 2005 could be of help:https://www.leavenetwork.org/fileadmin/user_upload/k_leavenetwork/annual_reviews/2005_annual_report.pdf).

-Since one of the limitations of the study is that individual sociodemographic information from 2005 is used as a proxy for the sociodemographic information at the time of childbirth, it would be advisable to clarify how much time, on average, had elapsed from birth to 2005.

-There could be unmeasured cultural/normative explanations that might account for some of the findings (for instance, actual norms in society, beyond the policy sphere about the importance of breastfeeding and support for relatively prolonged breastfeeding/intensive mothering ideals). Perhaps the cross-national qualitative work by Grunow and Evertsson 2016 ("Couples' transitions to parenthood") could provide some insights on this matter, although it covers a more recent time period than the one analyzed. It could also be the case that, besides the social policies considered regarding parental leave and the enhancement of breastfeeding, some countries also provide better support than others for breastfeeding mothers at the primary health care level. I would recommend paying some attention to these possibilities in the discussion.

Author Response

Response to reviewer 2

We appreciate the time you have taken to read our work and the suggestions you have made. Below, we explain how your remarks were addressed throughout the manuscript. All review changes are highlighted throughout the manuscript.

The manuscript addresses an important and so far not sufficiently explored question, and does so in a systematic, thorough and methodologically sound manner. It uses a multilevel design that makes it possible to carry out an analysis including 21 European countries and a variety of policy regimes. Its findings are rich and conclusions fill several gaps in the existing literature. For these reasons, I consider the article should be published. I would nonetheless recommend considering some minor questions:

-Although the manuscript does make an important contribution to earlier research, greater emphasis could be made on both its scientifical and social relevance in the background section, so that it is quickly conveyed to the reader (for instance, one strength of the article that is not sufficiently underscored is that it actually contributes to different streams of research/disciplines/areas of study: the literature on breastfeeding/public health, earlier research on parental leave and policy effects, the socioeconomic determinants of health-related behaviors, research on parenthood behavior, etc.). 

  • We’ve rephrased the last part of the introduction to make our contribution much clearer. Both the scientific and social contributions have been highlighted in the introduction.

-The manuscript considers the relevant literature on the topic. Nevertheless, its theoretical framing (leading to the hypotheses formulated) could be strengthened. The exact expected theoretical mechanisms leading to each of the hypotheses should be spelled out more clearly. It could be useful to shorten the background section somewhat and instead expand the "study aim and hypotheses section" with more explicit explanations of the mechanisms expected to be at play in the different types of policy regimes. 

  • We’ve added some additional explanations to the ‘study aim and hypotheses’ section to make the mechanism of leave entitlements and national policies affecting breastfeeding behavior clearer, without overextending or repeating ourselves in this part of the manuscript.

-It should be clarified in the background section that the classification of countries in the typology used refers to their policies in 2005. It should be clarified how much time had elapsed, on average, between the date of birth and the date of the survey.

  • We made sure this is especially clear in the manuscript by emphasizing this once again. Regarding the time lapse between the date of birth and the date of the survey: because of the phrasing of questions in the Eurobarometer we do not know when mothers gave birth. We realized this was problematic, and therefore constricted the time interval to not include women over the age of 40. We know that the majority of women become mothers no earlier than 18 years old, meaning there is a maximum of 22 year time difference. According to Eurostat, the average European woman becomes a mother at 30.6 years with the lowest average in Bulgaria of 25.1 years and the highest average in Greece of 33.8 years. In our methodology section, we added a part of this explanation so this is clear fur future readers.

- Parental leave in the male breadwinner model is claimed to be unpaid or paid at low, flat-rate levels, usually below 60% of the wage. Nevertheless, this is not always the case - Spain f.i. was an exception in 2005, with 100% paid maternity leave (yet with a maximum ceiling) during 16 weeks; even though longer parental leave was unpaid). It could be useful to check whether there were any other such exceptions 2005 (perhaps Moss & O'Brien 2005 could be of help: https://www.leavenetwork.org/fileadmin/user_upload/k_leavenetwork/annual_reviews/2005_annual_report.pdf).

  • I’ve checked the suggested report from the leavenetwork.org. We emphasized in our manuscript it is the parental leave that is usually unpaid, or paid at low, flat-rate levels. Maternity leave benefits are generally more generous throughout the European countries. For Spain, we found that this is likewise paid at a low flat-rate level for parental leave only.

-Since one of the limitations of the study is that individual sociodemographic information from 2005 is used as a proxy for the sociodemographic information at the time of childbirth, it would be advisable to clarify how much time, on average, had elapsed from birth to 2005.

  • As we’ve explained in the response above, this was unfortunately not an option. And we have addressed this concern in our methods section as well as the discussion section (limitations).

-There could be unmeasured cultural/normative explanations that might account for some of the findings (for instance, actual norms in society, beyond the policy sphere about the importance of breastfeeding and support for relatively prolonged breastfeeding/intensive mothering ideals). Perhaps the cross-national qualitative work by Grunow and Evertsson 2016 ("Couples' transitions to parenthood") could provide some insights on this matter, although it covers a more recent time period than the one analyzed. It could also be the case that, besides the social policies considered regarding parental leave and the enhancement of breastfeeding, some countries also provide better support than others for breastfeeding mothers at the primary health care level. I would recommend paying some attention to these possibilities in the discussion.

  • Absolutely, we agree our work doesn’t cover the wide range of macro-level determinants. To address your concern, we’ve added some suggestions to our discussion section: c-section rates (delivery type), presence of female physicians, childcare services and benefits, societal expectation to breastfeed…

Thank you for taking the time to read our revised work.

Sincerely,

The authors

This manuscript is a resubmission of an earlier submission. The following is a list of the peer review reports and author responses from that submission.

Round 1

Reviewer 1 Report

Review “Motherhood in Europe. An examination of parental leave regulations and breastfeeding policy influences on breastfeeding initiation and duration

This article focuses on a relevant topic for the literature, that is, a European comparison on how parental leave schema may affect breastfeeding (initiation and duration). The text is well written and structured. The theoretical framework is appropriate, and the literature review is relevant, but it may need a more recent updated. The method is ambitious, and the statistical technique is well enough explained. The results are interesting, but some limitations arise from the ambitious proposal of the author. In general, I would recommend the article for publication on Social Sciences if the next comments are considered.

  • Sometimes is difficult to read since many acronyms are used. I would indicate that an acronym will be used in the rest of the article. For example, breastfeeding initiation (from now on BFI). This is particularly the case when some parental leave types are expressed in acronyms and other do not, even in tables (i.e. table 3).
  • One of the most controversial decision in this paper is to use a 15 years-old database. The author is aware and explain that is the only way to carry out a comparative analysis on so many countries and he/she expresses the limitations about this fact. However, in my opinion, it should be further described how was contextual situation in Europe in 2005. This period is quite different from the economic crisis period from 2008 or our current COVID situation in 2020. So, I think, that at least one paragraph could be used to inform about general European situation in 2005.
  • In the background section the author quotes researchers that warm about the limitation of comparative data (Cattaneo et al, 2005; Yngve and Sjostrom, 2011). And I wonder if the author may explain how he/she is going to solve these limitations.
  • When using Ciccia and Verloo’s typology, it would be relevant to know what it means by “paid at low, flat rate levels”. What is the percentage of the wage that is considered low?
  • The information of Graph 1 is located also in table 2. I do not know if it is appropriate to repeat this information. Also, if the publication is black and white the two lines are not differentiated.
  • Maybe the most important modification would be to develop further the parental leave scheme in the countries analysed, in 2005. For that, I think it is very recommendable to read and quote the International Leave Network reports (they can be found in this webpage: https://www.leavenetwork.org/annual-review-reports/)
  • Also, I would suggest using a more recent reference in page 16 (587-588) when it is stated that, the current leave policy (in 2012?) is mostly and expansion of leave time for mothers, fathers and parents, but rarely includes further flexibility measures (Ciccia and Verloo, 2012). In my opinion, this is not the case for 2020, many countries have rise the paternity leave or fathers’ quota, and also it has been create bonuses to promote that fathers take the leave alone when the mother return to work. I would suggest more reading about current situation of leave in Europe. Also, I would strongly recommend reading Koslowski, A., Blum, S., Dobrotic, I., Moss, P., & Macht, A. (2019). 15th International Review of Leave Policies and Related Research 2019. Retrieved from International Network on Leave Policies & Research Webpage: https://www.leavenetwork.org/fileadmin/user_upload/k_leavenetwork/annual_reviews/2019/2._2019_Compiled_Report_2019_0824-.pdf).
  • In page 22 (495-496), when it says that “working fewer hours in paid employment indicates higher breastfeeding rates”, is it only for initiation? It should be clarified.

Reviewer 2 Report

The manuscript 'Motherhood in Europe . An examination of parental leave regulations and breastfeeding policy influences on breastfeeding initiation and duration' is discussing the connection between policy-level national differences and breastfeeding. It is an important paper, addressing a topic that has not been thoroughly studied previously.

The paper relies on the Eurobarometer 2005 data, which offers some intriguing directions of analysis, connecting variables that have previously been studied only on the personal level with national-level variables. In this, the manuscript adds an important contribution to understanding the connection between policy and breastfeeding, and more generally – of the relationship between personal behavior and policy.

However, the manuscript suffers from several structural problems, which undermine its potential for contribution, and should be addressed before it is ready for publication. While the data is presented quite clearly and comprehensively in the results section, the other parts of the manuscript are less well organized, which undermines its potential contribution.

First, The authors fail to state their hypotheses clearly. While the authors refer to their 'expectations' in several places in the literature review (p. 5 lines 207-8; p. 6 lines 260-1), they fail to provide an organized, exhaustive list of hypotheses. The absence of hypotheses also reflects in the discussion sections, which does not answer the most basic question for a qualitative study – were the research hypotheses confirmed?

Likewise, I find the organization of the discussion section very confusing. Instead of discussing the (missing) hypotheses, this section relates to three findings – personal-level determinants of breastfeeding, connections between leaves and breastfeeding, and between breastfeeding-related policies and breastfeeding. However, this discussion is very unorganized and, as a result, hard to understand. Some examples:

  • On page 15 (line 494) the authors claim that their research does not support the time-availability argument – however, this argument has not been discussed earlier, in the literature review, and is introduced here for the first time
  • Similarly, page 16 offers a discussion on the connection between the 'baby-friendly concept' and the variety of maternal and baby health constellation across Europe (lines 545-553) – this discussion should appear in the literature review.
  • On page 15 (lines 511-518), the authors discuss the Danish parental leave and its uniqueness in their sample. The relevance of this discussion is not clear to me, but if the authors believe it to be necessary, the appropriate place for it is in the presentation of the parental leave typology in the literature review.
  • On page 16, the authors turn to examine the paper's limitations – why is this not in the conclusions section?

A difficulty I found with the literature review is the author's reliance on Ciccia and Verloo's (2012) model. While this model is a solid choice, I did not find the theoretical reasoning for this choice sufficient. There are a plethora of typologic models of family policy and specifically parental leave regimes (see, for example, Saraceno, 2018). I would expect to see a discussion on the specific choice of model.

Furthermore, the choice of using a regime typology instead of referring to specific policies, while theoretically grounded, could also be empirically tested. Considering the findings, which do not find a strong connection between regime type and breastfeeding choices, the authors could test an additional statistical model in the analysis section, relating to specific policies and not to regimes. Testing parameters such as leave length, replacement rate, etc. could provide interesting re

Some additional minor points:

  • The title contains a period – is this intentional?
  • The use of acronym (BFI, BFD, BFH) came out very confusing for me, as they are very similar. I would consider using alternatives.
  • The organization of the literature review is not clear to me. The authors discuss several models of connections between personal and policy levels (Risman's model, Bently at al's model, and more) and do so in detail – but these models are not used later to explain the findings. While a discussion on the relationships between personal behavior and policy is certainly in place here, the level of detail is distracting.
  • While describing Risman's model (p2), the authors list values on the structural level, while policies are listed on the cultural level. This seems like a mistake as they are usually attributed reversely – policy as structural, values as cultural. If this is not a mistake, the authors should explain (and substantiate) their choice (or drop it altogether, as per my previous comment).
  • Page 3 line 126 – the authors refer to Fraser's 'study' – the cited work is a theoretical paper
  • Page 4, line 169 – the authors state, 'We identify two versions of [Ciccia and Verloo's] model' – is this an original distinction or taken from Ciccia and Verloo's work? If the last, attribute correctly; if the first, some empirical work is needed to substantiate it (or stay with Ciccia and Verloo's work, as a regime typology is outside the scope of this paper)
  • While the authors relate to their inability to infer about causality (p. 17 line #602), they imply causality at several points in the discussion (e.g., "International Code, however, negatively influences BFI and BFD" – p. 16 line #562; and more). This not merely technical – the results may be highly influenced by mitigating variables, or the direction of causality may be opposite (countries with low levels of BFI being more eager to implement the international code, for example)
  • On page 15, line 519, the authors state that 'only the limited caregiver model countries have higher duration rates,' but it is not clear higher than what.

To summarize – I believe this paper has the potential for making a good contribution to the literature. However, the aforementioned problems should be addressed before it is fitting for publication.

Reviewer 3 Report

The paper is not well focused on a single hypothesis but addresses multiple issues (social, political, health) at once. It feels like it could almost be two or three manuscripts because the aspects around the social concepts and the results from the breastfeeding data, as well as other data on policies and the Itnernational Code are not well woven together well.

The paper becomes particularly problematic when the authors interpret their exploratory findings as casual. Cause and effect cannot be determined from this type of observational, cross-sectional study, which compares individual-level data with meta-level (national programme and policy) data. There is too little reflection on what seems to be counter-intuitive findings (global policies and programs aiming to protect and support breastfeeding are reported by the authors to negatively impact breastfeeding).

Abstract: The authors state “We find that the International Code is negatively associated with initiation and duration. BFHs have no effect on initiation or duration.” These conclusions both overinterpret and misinterpret the data. The main research question which is discussed substantively in the text (“how countries’ social policies and mothers’ individual social positions shape breastfeeding outcomes”) is not mentioned in the abstract.

Please define BFH or spell-out the abbreviation BFH in abstract and the paper.

Introduction: This section needs to clearly state the authors’ hypothesis.

Line 25: According to the data presented in Graph 1, the breastfeeding initiation rate in Lithuania is less than 80 percent, which is lower than other countries surveyed, i.e. not “high.”

Line 29: Suggest to spell out BFD abbreviation at first use

Line 54: There is a problem with the citation manager: "Hodgson, 1986 #322"

Line 76, Figure 1: It is not common to provide figures in the Introduction section. Please consider if this is better suited for later in the paper, if at all. The social ecological model is generally well-known amongst readers of this kind of journal. If you keep it, please cite the original source, it is much older than 2003.

Lines 83-89: Breastfeeding rates in Sweden are discussed based on a report that is almost 20 years old, but there is newer data available.

Line 88, lines 94-95: Current references are needed when discussing the inconsistencies in reporting breastfeeding statistics across European countries.

Line 90: Please explain who are “key personnel”?

Line 98: Please clarify "low family spending.” Also, how did the report explain the relationship between a high percentage of women in parliament and breastfeeding rates?

Lines 149 -162: “the male breadwinner model” Is this a formal name? If so, please cite and define it in methods. Here the authors listed Western and Eastern European countries, but there are potentially different roles for masculinity/fathers across these regions in Europe. Perhaps elaborate in the discussion. “The limited caregiver model” Is this the formal name? If so, please cite and define in methods. “The caregiver parity model” Is this the formal name? Please cite and define in methods. What is meant exactly by “generous?”

Table 1: The Ten Steps do not need to be included in a table; it is sufficient to cite them. They were updated in 2018, please update this in the citations accordingly.

223: the Baby-friendly Hospital Initiative is not a policy but a programme

Please use in short form “the Code” or the “the International Code,” but not both.

238-240: The connection between “precarious social backgrounds” and baby-friendly hospital accreditation does not make sense.

240-250: This statement is incorrect, “the legal consequences of implementing or failing to implement the Code.” The International Code is a set of recommendations, not a legal framework or a law. The International Code itself does not stipulate “legal consequences.”

254-256: “These studies reveal the occurrence of severe violations of the Code in countries in which it has not been legally implemented; we expect this situation to be similar in European countries where infant feeding marketing is pervasive.” Yes, it is the same in Europe. There is current literature on this, which should be added.

Line 267: legislative frameworks were not evaluated in this study

Methods:

274: In the Methods section, please define all of the breastfeeding terminology used like “breastfeeding initiation.” Does “breastfeeding duration” mean any breastfeeding at all?

Please give references or more explanation of the 8-point scale measuring International Code implementation

Line 278: The authors report 1,000 respondents per country, which would mean 21,000 respondents. The Eurobarometer data described for this paper used data from only 2549 mothers from 21 countries (This is only 0.12% of 21,000 respondents or an average of 121 women from each country). This cannot be representative? Please give more details on the distribution of mothers in each country included.

Lines 287: “Papacostas and Soufflot de Magny, 2012” The author(s) refer to the Eurobarometer 2005 but do not cite it in the references. Is this data publicly available?

288: Please define “breastfeeding measure.” Why is this study supposedly the only survey that allows European cross-national comparisons?

Line 228: “Partly”

Lines 320-321: Difference between unemployed and non-employed?

Line 322: Strange use of citation and parentheses.

Lines 391-400: Based on what descriptive table? Without any explanation of what is meant, the authors begin to discuss “degree of implementation of the Code.” Later, in Table 2 it is referred to as an “index score.” It is not clear what, if anything, these values are calculated or based upon. Is this an index created by the authors themselves?

Lines 417-421: “Higher educated women 417 are more likely to initiate breastfeeding (OR = 1.034)” Is this the correct interpretation with an OR of 1.03? “Women with a partner are more likely to initiate breastfeeding (OR = 1.060).” Is this the correct interpretation with an OR of 1.06?

Table 2: Please provide number of respondents (n) not just %. Please provide number of hospitals (n) not just %. Where does BFH data come from? For the index score and BFH, what is the number in parentheses? Please provide footnotes to explain abbreviations.

Table 3: What is “cons?” Please give a footnote to explain all abbreviations in the table. For column “Model 3,” the variable “International Code” cannot be statistically significant if the confidence interval was [0.531-1.700]

Results: This section begins loads of interpretation of the results which is best done in the discussion section.

Discussion: How do the rates reported in Graph 1 relate to other published literature like that from Lubold in 2017?

Line 580: suggest to name this the Limitations section. Please discuss other limitations of the Eurobarometer data, besides the fact that it is a bit old.

Lines 429-431: “We found that countries with a higher degree of International Code implementation show lower levels of BFI (OR = 0.830) than countries with a lower degree or no degree of implementation.” Is this perhaps because countries have implemented policies as a result of low BF rates (i.e. reverse causation?) Please discuss this possibility of confounding. It would also be helpful to know a timeline-- when the countries implemented the International Code compared to when these breastfeeding data were collected in 2005.

Conclusions: The conclusion sections glosses over the vastly different conclusions made in the abstract.

References: The references provided are too old, the majority are more than 15 years old (>30 references are before 2005). The text and literature should be updated to include more recent references from Europe.